# Active suppression prevents the return of threat memory in humans

Ye Wang [1], Zijian Zhu[2], Jingchu Hu[3], Daniela Schiller[4] & Jian Li [1,5]✉

Unbidden distressing memories inflict serious damage on mental health. Extant research highlights the importance of associative learning in modulating aversive memory. We report that conscious active suppression eliminates learned fear responses independent of memory triggers and is related to individual difference in thought control ability; in contrast, thought diversion only reduces cue-specific fear response. These results suggest potential avenues for treatment of persistent maladaptive memories by engaging declarative mnemonic control mechanisms.

[1] School of Psychological and Cognitive Sciences and Beijing Key Laboratory of Behavior and Mental Health, Peking University, Beijing, China. [2] School of Psychology, Shaanxi Normal University, Xi'an, China. [3] Department of Anxiety Disorders, Shenzhen Kangning Hospital, Shenzhen Mental Health Institute, Shenzhen, China. [4] Departments of Psychiatry and Neuroscience, and Friedman Brain Institute, Icahn School of Medicine at Mount Sinai, New York, NY, USA. [5] PKU-IDG/McGovern Institute for Brain Research, Peking University, Beijing, China. ✉email: leekin@gmail.com

Not all memories are precious. Making matters worse, intrusive traumatic memories can even be triggered by mundane events in real life. Therefore, understanding how to prevent the return of aversive memory has been one of the fundamental research topics in psychiatry and clinical psychology. Associative memories are learned through direct experience such as Pavlovian conditioning and usually require very few learning trials to remain robust and persistent.

To prevent the return of aversive associative memories, several techniques have been developed in recent literatures that include extinction[1] (the repeated non-reinforced presentation of conditioned stimuli), reappraisal[2] (top-down cognitive regulation) and reconsolidation[3] (the process of re-stabilizing a reactivated memory). For example, both pharmacological and behavioral interventions targeting reconsolidation processes have shown promise to alter or even erase unwanted memories in both rodents and humans since reactivation places memory into a labile state and renders it susceptible to interference[4–8].

However, aversive memories are typically encoded under specific context and associated with multiple triggering sensory cues, and can be long consolidated before the memory modulation efforts. In Pavlovian aversive conditioning paradigm, for instance, modifying the contingency between a specific conditional stimulus (CS) and unconditional stimulus (US) pairing during the reconsolidation time window only specifically disrupts the reactivated CS–US memory trace, while leaving other memory traces of the same US intact[7,9,10].

Research on other memory types, such as declarative memory, has highlighted the importance of cognitive suppression on memory recall[11,12]. In the influential think/no-think (TNT) episodic memory paradigm, when participants encounter memory triggering cues and actively keep unwanted cue-associated memory out of awareness (no-think), they are less capable to recall the rejected memory later, even under incentives for accurate recall[11]. This amnesia effect is believed to be caused by the memory suppression itself via erasing the original memory engram, as demonstrated by the cue-independent recall deficit from the independent probe test, and has distinct behavioral and neurobiological mechanisms[12,13].

Other mnemonic control methods, such as thought diversion, are based on interference theory and suggest that specific target memory can be interrupted during recall if there are multiple cue-associated memories competing for the cue-triggered memory retrieval. For example, in memory interference studies, previously established cue-target memory recall was impaired when the participants were asked to associate the cue with a substitute memory target[13–16]. However, compared to direct memory suppression, memory interference effect is believed to be cue-specific and cannot generalize to other target memory associated cues[13,15,16]. Moreover, people with higher self-reported thought-control capabilities show more severe cue-independent recall deficit, suggesting that the direct suppression processes are associated with individual difference in control abilities over intrusive thoughts[17].

Despite the popularity of the TNT and memory interference paradigms in declarative memory research, it is yet unclear whether active suppression and interference of memory retrieval can be applied to other memory types such as associative threat memory, and, if so, whether this forgetting effect is related to thought-control ability[17]. It further begs the question of whether memory retrieval suppression (active suppression) and memory interference (thought diversion), both of which widely studied in the mnemonic control research, work similarly on conditioned threat memories.

Here we tested whether conscious memory suppression during threat memory extinction training after the establishment of CS (neutral visual stimulus)–US (mild electric shock) conditioning would diminish the return of fear response, measured by the fear memory reinstatement effect. To assess the generality of memory suppression effect, we also tested the effect of thought diversion, a cognitive process to mentally associate CS with a new neutral outcome (see Methods), on later fear return.

We hypothesized that similar to declarative memory research, direct memory suppression would diminish the return of associative threat memory, irrespective of the triggering CS, whereas thought diversion only eliminates CS-specific fear threat response. If this is true, we would also expect that individual differences in thought-control ability, a measure of individual's capability to control unwanted memory intrusion, would be associated with the forgetting effect induced by direct memory suppression, but not thought diversion.

## Results

To test these hypotheses, we designed two experiments by combining the active suppression manipulation and thought diversion manipulation with fear extinction training[11,13]. We adopted a double-cue paradigm to assess the recovery of fear threat memories (Fig. 1a, b). Both experiments lasted 2 days and included three phases: fear acquisition (Day 1), fear extinction combined with direct active suppression (actively expel what might have followed conditioned stimuli out of awareness) for Experiment 1 and with thought diversion (mentally relate conditioned stimuli to neutral and soothing natural scenes) for Experiment 2 (Day 2), and a test phase after fear reinstatement (30 min after extinction training on Day 2) (see Methods for details).

In our experiments, participants were trained to associate two colored squares on the computer screen (conditioned stimuli (CS+)) with electric shocks in the acquisition phase (i.e., CS1+ and CS2+). During the extinction phase, active suppression (Experiment 1) or thought diversion (Experiment 2) was applied to one of the CS+ and fear responses to all the CS were examined in the test phase to test the effects of different mnemonic control methods. Therefore, both experiments can be designated into three phases: acquisition, extinction, and test (Fig. 1).

All the conditioned fear responses were measured using the spontaneous skin conductance responses (SCR). Participants of both experiments also completed the Thought Control Ability Questionnaire (TCAQ) to assess their perceived control abilities over intrusive thoughts[18]. Therefore, the only difference between Experiments 1 and 2 is the mnemonic control strategies: active suppression or thought diversion, respectively.

We observed that CS1+ and CS2+ elicited similar conditioned responses on Day 1 in both experiments. A mixed-model three-way ANOVA showed a significant main effect of trial ($F_{9,477} = 5.311$, $P < 0.001$, $\eta^2 = 0.091$), but no effect of experiment (suppression vs. diversion), CS+ (CS1+ vs. CS2+, defined as the mean SCR differences between CS+ and CS–) or their interactions (all $P$ values > 0.1). Post-hoc $t$-tests showed that the mean responses of CS1+ and CS2+ were significantly higher than those of CS– in both experiments (CS1+: $t_{27} = 7.612$, $P < 0.001$; CS2+: $t_{27} = 10.04$, $P < 0.001$ for Experiment 1 and CS1+: $t_{26} = 7.174$, $P < 0.001$; CS2+: $t_{26} = 7.645$, $P < 0.001$ for Experiment 2), suggesting successful fear acquisition (Fig. 2a, b). To examine whether participants in both experiments achieved similar levels of acquisition, we conducted a 2 (experiments) × 2 (CS+) mixed-model two-way ANOVA on the late phase of fear acquisition (last five trials), which showed no effect of experiment ($F_{1,53} = 1.819$, $P = 0.183$, $\eta^2 = 0.033$), CS+ (CS1+ vs. CS2+; $F_{1,53} = 0.297$, $P = 0.588$, $\eta^2 = 0.006$) or their interaction ($F_{1,53} = 2.361$, $P = 0.134$, $\eta^2 = 0.042$). Importantly, the mean differential SCR responses induced by CS1+ and CS2+ of the late phase of fear acquisition

## a

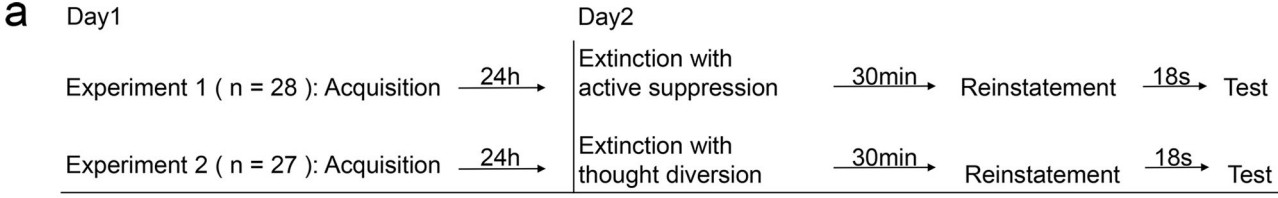

## b

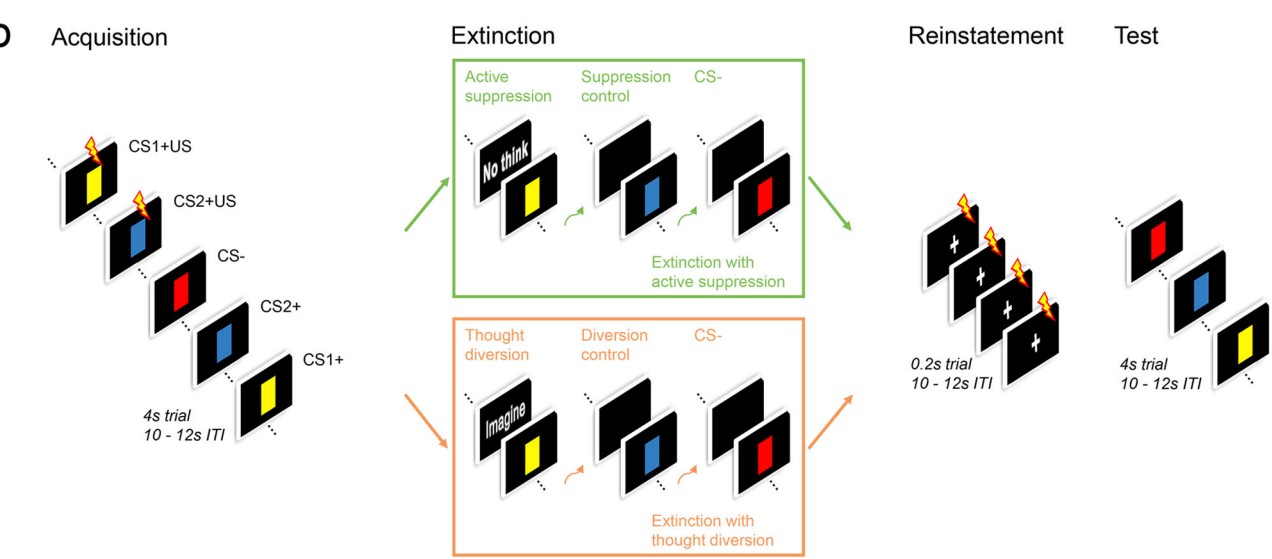

**Fig. 1 Experimental design and procedure. a** Experimental paradigm timeline. **b** Participants underwent fear acquisition, extinction, reinstatement, and test phases in both experiments. In Experiment 1, extinction training was coupled with active suppression only on CS1+, but not CS2+ (suppression control) or CS− (upper green rectangle), whereas in Experiment 2, extinction was combined with thought diversion treatment on CS1+ but not CS2+ (diversion control) or CS− (lower orange rectangle).

(CS1+ vs. CS2+, defined as the mean SCR differences between CS+ and CS−) were not different in both experiments ($t_{27} = -1.439$, $P = 0.162$; $t_{26} = 0.703$, $P = 0.488$, respectively; Fig. 2c).

We then set out to examine whether participants underwent successful extinction training on Day 2. The similar three-way ANOVA showed a significant effect of trial ($F_{9,477} = 9.742$, $P < 0.001$, $\eta^2 = 0.155$) and experiments ($F_{1,53} = 4.161$, $P = 0.046$, $\eta^2 = 0.073$), but no significant effect of CS+ or interactions (both $P$ values > 0.05). To test whether participants in both experiments achieved similar levels of extinction, we ran a 2 (experiments) × 3 (CS1+, CS2+, and CS−) mixed-model two-way ANOVA on the last trial of extinction phase and found that there was no main effect of experiment ($F_{1,53} = 0.001$, $P = 0.996$, $\eta^2 = 0.001$), across CS (CS1+, CS2+, and CS−; $F_{2,106} = 0.472$, $P = 0.625$, $\eta^2 = 0.009$) or interaction ($F_{2,106} = 1.299$, $P = 0.277$, $\eta^2 = 0.024$). Post-hoc $t$-tests further confirmed that there were no significant differences between CS+ and CS− responses (all $P$ values > 0.1) in both experiments (Fig. 2c). Similarly, the differential SCR responses related to CS1+ and CS2+ on the last trial of extinction (CS1+ vs. CS2+, defined as the mean SCR differences between CS+ and CS−) were not different in both experiments ($t_{27} = -0.469$, $P = 0.643$; $t_{26} = 0.832$, $P = 0.413$, respectively; Fig. 2c).

Fear reduction was assessed again using a three-way ANOVA with the within-subjects factors CS+ (CS1+ vs. CS2+, defined by the mean SCR differences between CS+ and CS−), phase (late phase of acquisition vs. last trial of extinction), and between-subjects factor experiment (suppression vs. diversion). There was a significant effect of phase ($F_{1,53} = 96.807$, $P < 0.001$, $\eta^2 = 0.646$), but no effect of experiment, CS+ or their interactions (all $P$ values > 0.05). Post-hoc $t$-tests showed that there were significant decreases in fear response from acquisition to extinction of both experiments

(CS1+: $t_{27} = 5.519$, $P < 0.001$; CS2+: $t_{27} = 6.168$, $P < 0.001$ for Experiment 1 and CS1+: $t_{26} = 6.699$, $P < 0.001$; CS2+: $t_{26} = 6.308$, $P < 0.001$ for Experiment 2).

Inspired by the declarative memory literature, we hypothesize that direct suppression should cause cue-independent amnesia of learned fear, whereas thought diversion's effect is cue-specific[11,13]. To test this, participants went through a reinstatement treatment via exposure to the aversive stimuli (US) before the test phase[6,9,19]. Successful reinstatement effect can be illustrated by the differential CS+ (compared to CS−) related SCR responses from the last trial in the extinction phase to the first trial in the test phase. Consistent with our hypothesis, reinstatement effect is stronger in Experiment 2 (diversion) than in Experiment 1 (direct suppression) (experiment × phase (extinction vs. test), $F_{1,53} = 14.57$, $P < 0.001$, $\eta^2 = 0.216$; see Methods for details). More specifically, in Experiment 1, direct fear memory suppression following CS1+ in the extinction training eliminated fear reinstatement for both CS1+ and CS2+ since the SCRs related to both CS+ were not different from those of the CS− ($t_{27} = -1.844$, $P = 0.076$; $t_{27} = -1.333$, $P = 0.194$) in the test phase (Fig. 2c), whereas in Experiment 2, thought diversion on CS1+ only successfully eliminated the fear reinstatement for CS1+ (compared to CS−, $t_{26} = -0.992$, $P = 0.33$), while leaving the fear SCR response to CS2+ intact ($t_{26} = -4.663$, $P < 0.001$; Fig. 2c). Together, these results suggest cue-independent and cue-specific effects of direct memory suppression and memory diversion, respectively (Fig. 2c).

In line with our second hypothesis, thought-control abilities only affect the level of fear recovery in Experiment 1 but not Experiment 2 (experiment × thought-control ability interaction, $F_{1,51} = 6.744$, $P = 0.012$, $\eta^2 = 0.117$). In addition, participants with higher TCAQ scores showed greater diminishment of fear

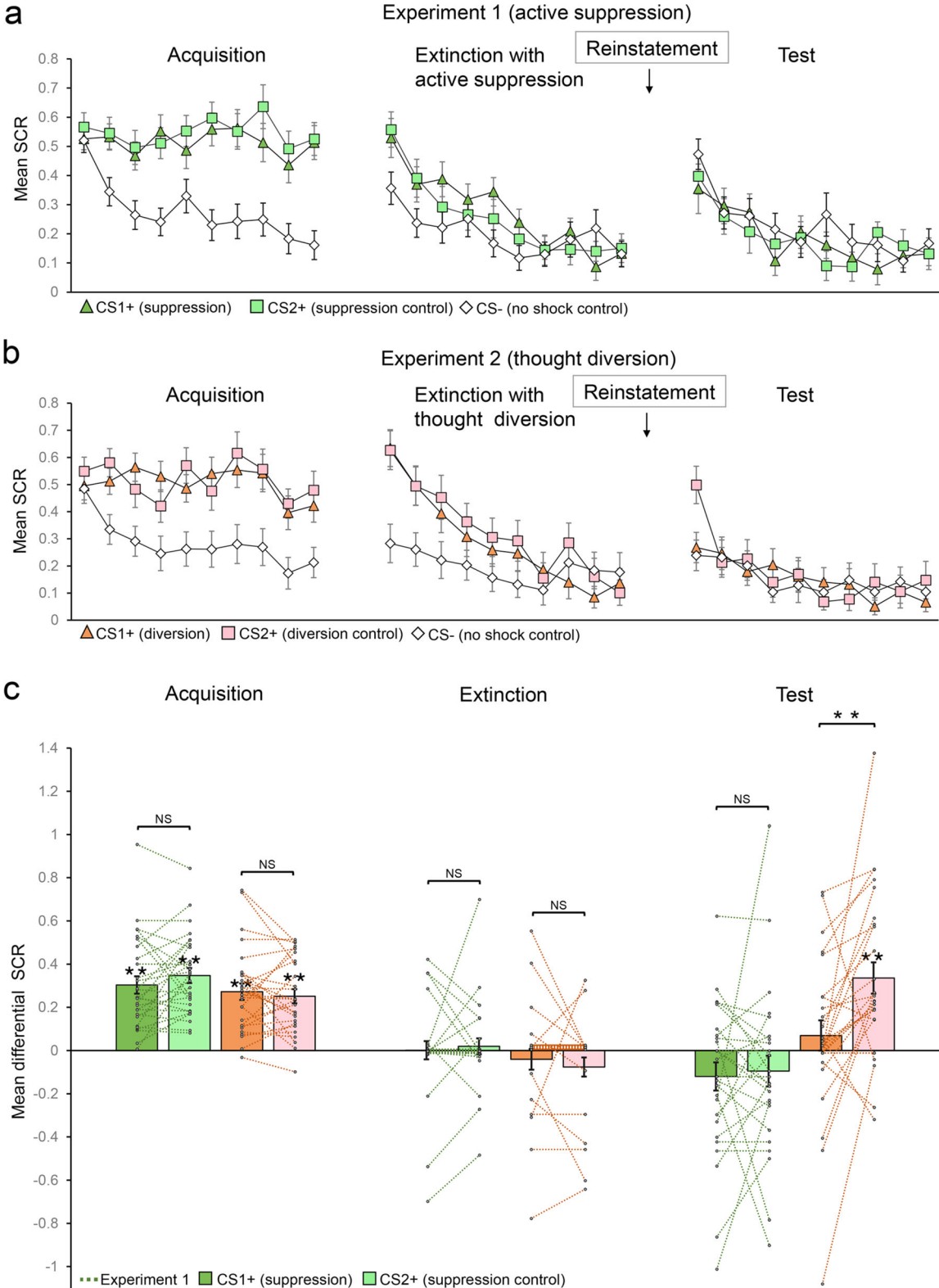

**Fig. 2 SCR responses to conditioned stimuli in both experiments. a** Mean SCRs of fear conditioned stimuli CS1+ (suppression) and CS2+ (suppression control), and the no-shock control stimulus (CS−) across fear acquisition, extinction, and test phases in Experiment 1 (active suppression). **b** Mean SCRs of fear conditioned stimuli CS1+ (diversion) and CS2+ (diversion control), and the no-shock control stimulus (CS−) in Experiment 2 (thought diversion). **c** Mean differential SCRs (CS1+ minus CS−, and CS2+ minus CS−) in acquisition (late phase), extinction (last trial), and test phases (plotted as fear recovery index: the difference between the first test trial and the last extinction trial for CS1+ and CS2+ of both experiments). **P < 0.001. NS non-significant. Error bars represent standard errors.

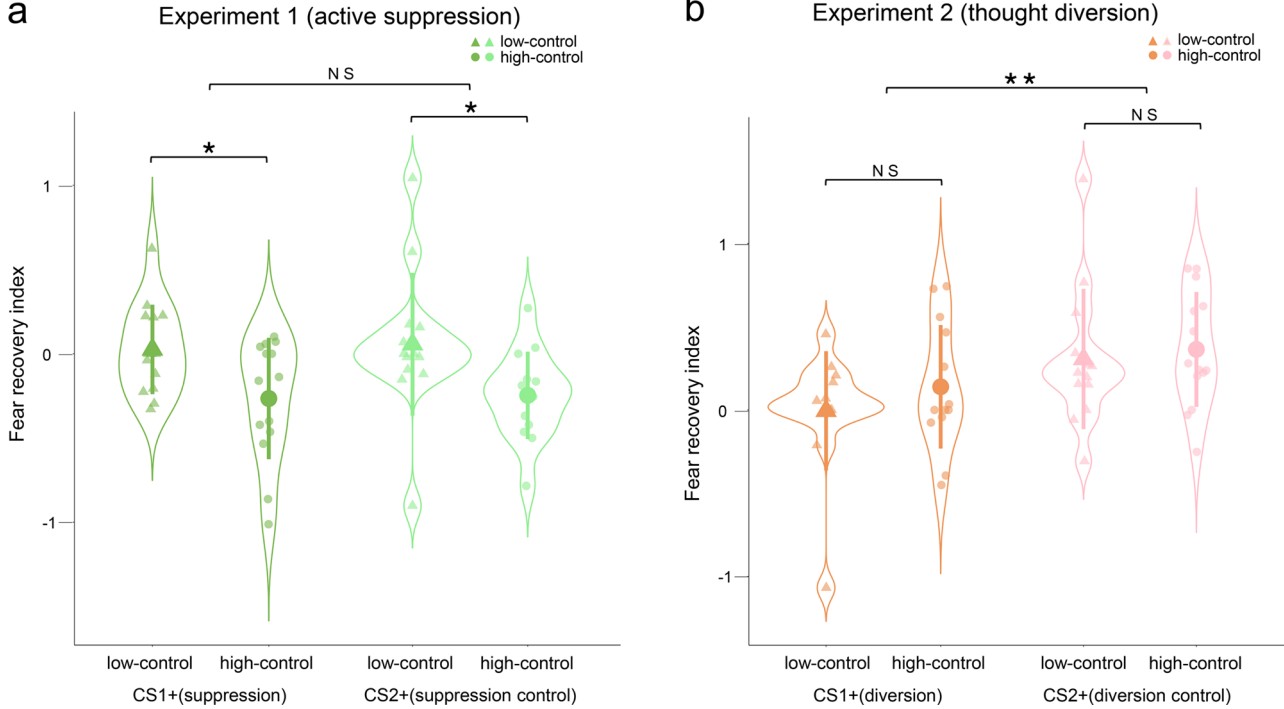

**Fig. 3 Fear recovery as a function of thought-control abilities. a** After direct active suppression manipulation, high thought-control participants showed less fear recovery than low-control participants for both CS. **b** Thought diversion only diminished the CS-specific fear response, and thought-control ability was not associated with the fear recovery of CS+. For both experiments, means of fear recovery index are depicted as large circles for high-control subjects and triangles for low-control participants. Standard deviations are depicted in bars, measures for each subject in small dots and triangles, and distribution in violin shape. *$P < 0.05$, **$P < 0.001$. NS non-significant.

responses for CS1+ and CS2+ after direct suppression in Experiment 1 (thought-control ability; $F_{1,26} = 9.017$, $P = 0.006$, $\eta^2 = 0.258$; see Methods for details; Fig. 3a). Interestingly, there was no difference of fear recovery between groups of high and low thought-control ability after thought diversion manipulation in Experiment 2 (thought-control ability; $F_{1,25} = 0.736$, $P = 0.399$, $\eta^2 = 0.029$; Fig. 3b). Importantly, there was no difference between high and low thought-control ability participants at the end of fear acquisition and extinction training in both experiments (all $P$ values > 0.10), suggesting that thought-control ability is specifically associated with the levels of fear reinstatement effect.

## Discussion

In two experiments with the double-cue fear learning, extinction, and reinstatement paradigm, we tested the effects of direct active suppression and thought diversion, two widely used mnemonic control methods in declarative memory research, on the return of conditioned threat memory response. It has been shown in the declarative memory research that executive control, via the engagement of dorsolateral prefrontal cortex (dlPFC) activities, leads to reduced hippocampal activation and impairs memory retention[11]. More importantly, this memory suppression approach seems to act upon the memory engram itself, rather than disrupt memory retrieval[11,20]. However, whether such a mechanism can also be applied to other memory types remains unknown.

The issue is further complicated by the fact that in declarative memory research, mnemonic control methods were typically tested on memories that had not gone through consolidation process[11,13,20] (usually less than 1 h after the training of cue-target memory association in the laboratory experiments). However, fear-related associative memory poses a greater challenge since it is usually well established before the memory modulation attempt such as in posttraumatic stress disorder

(PTSD) or phobia patients[6,7,9,21,22]. Therefore, it remains to be tested whether declarative memory inhibition (active suppression) and interference (thought diversion) methods can be applied successfully to the consolidated fear memory.

Here in two experiments, we showed that active memory suppression eliminated threat-related physiological responses in the test phase after fear reinstatement and this effect further generalized to memory traces that did not undergo suppression treatment. In contrast, thought diversion's effect on the return of fear was cue specific. It is worth noting that both active suppression and thought diversion were applied to the consolidated fear memory that was established 24 h before the extinction training. Also, although fear reinstatement is a robust phenomenon in fear learning literature, our interpretation that active memory suppression eliminated fear response hinged on the results in Experiment 2, where thought diversion was only effective on the CS+ to which it was specifically applied. The other CS+ showed the stereotypical fear reinstatement effect. These results were further corroborated by the findings that individual differences in thought-control ability were associated with the effect of memory suppression, but not thought diversion.

Previous literature suggest that there is a negative association between suppression-induced forgetting and thought-control ability[22,23]. Such observation corroborates well with reported suppression-induced forgetting impairment in PTSD or anxiety patients, who are characterized by deficient memory and intrusive thought control[24]. Our results resonate with these findings and further show that fear reinstatement reduction caused by active suppression is also sensitive to the thought-control ability whereas thought diversion induced forgetting is not (Fig. 3).

Together, these results suggest that direct suppression, by actively pushing unwanted memory out of awareness, might serve as a general mechanism in regulating memory retention across

different memory types. Indeed, previous research has shown that suppressing anticipatory dreadful events modulates activities in the ventromedial prefrontal cortex (vmPFC), a critical brain region also involved in fear extinction and reversal[25,26]. Therefore, it is possible that the interactions of dlPFC with brain structures such as vmPFC and amygdala may underlie the direct suppression effect on the return of threat memory observed in our experiments[1,27].

This research also speaks directly to the predominant view that reconsolidation might be the main target in modulating aversive memory[5–7,9,19]. In our task, fear reinstatement effect was tested 30 min after the fear extinction training where participants underwent active suppression and thought diversion treatments and significant threat memory impairment was detected. However, memory reconsolidation is thought to involve protein synthesis and usually take hours to complete, rendering it an unlikely mechanism for the memory deficits we observed[4,9]. Although future studies are needed to test how long the fear amnesia induced by direct suppression might last, our results nevertheless suggest that systemic suppression may engage distinct cognitive and neural mechanisms compared to memory reconsolidation and they could open new avenues to the understanding and treatment of psychiatric disorders caused by excessive fear or anxiety.

## Methods

**Participants**. All participants were students recruited from Peking University. They were right-handed with normal vision and had not participated in electric shock-related experiments before. All participants provided informed consent and were paid for their participation. This study was approved by the ethical committee of Peking University.

We conducted a power analysis[28,29] (G*Power) to determine the number of participants sufficient to detect a reliable effect. Based on average small-to-medium effect size of reinstatement effect between treatment group and control group on fear memories reported in the previous literatures (median $\eta^2 = 0.19$)[6,9,30], a total of 48 participants for both experiments were needed to detect a significant effect ($\alpha = 0.05$, $\beta = 0.8$, 2 (experiments) × 2 (phases) × 2 (CS) three-way ANOVA interaction effect).

Due to the well-known high attrition rate of SCR data collection[7,31,32] in recent fear learning experiments, we first recruited a total of 90 human subjects (44 females; mean age = 22.2, SD = 2.59). Thirty-five participants (15 females; mean age = 22.4, SD = 2.76) who were either "non-responders" (no SCR response to any CS) or "non-learners" during fear acquisition or extinction[7,10,31,33,34] were excluded from further analysis: Nineteen participants were excluded after Day 1 of testing for not responding toward non-reinforced CS (CS1+, CS2+, or CS–) (non-responders, $n = 9$ and 10; Experiments 1 and 2, respectively). Four participants ($n = 3$ and 1, respectively) failed to show the evidence of fear acquisition since their CS+ responses were numerically smaller than the CS– responses in the latter half trials of acquisition and the difference between the CS+ response and the CS– response also decreased during acquisition (non-learners). Finally, 12 participants ($n = 8$ and 4, respectively) failed to show the evidence of fear extinction on Day 2, since their CS+ responses were larger than the CS– responses in both the last trial and the latter half trials of extinction and the difference between the CS+ and the CS– responses increased during fear extinction (non-extinctioners).

The criterion for SCR difference (CS+ vs. CS–) increase during extinction was defined as the conjunction of the following conditions: the SCR difference of the last trial larger than that of the first trial of extinction, and the mean difference of latter half trials larger than that of the first half trials in the extinction, and the mean difference of latter half trials larger than that of the first trial, and the last trial difference larger than the mean difference of the first half trials in the extinction[7,31,32].

Therefore, our final sample included a total of 55 participants: 28 healthy participants (15 females; mean age = 22.27, SD = 2.64) in Experiment 1 and 27 participants (14 females; mean age = 22.12; SD = 2.29) in Experiment 2. It should be noted that our main results remain robust after including the 4 non-learners and 12 non-extinctioners into the statistical analyses.

**Stimulus and psychophysiological stimulation**. For both experiments, three squares with different colors (yellow, red, and blue) were used as CS1+, CS2+, and CS–, respectively. During fear acquisition phase, CS1+ and CS2+ trials (16 each) were paired with electric shocks (US) on a 37.5% partial reinforcement scheme, and the CS– (10 trials) was not paired with shocks. The first trial of acquisition, extinction, and test is a dummy stimulus (green cylinder) to account for the initial orienting response and not included in the analysis[33,34]. A pseudo-random

stimulus order was generated for acquisition and extinction phases of both experiments. In the test phase, to exclude the possibility that the difference between CS1+ and CS2+ was simply caused by the presentation sequence of CS1+ and CS2+, half of the participants completed the test phase using a pseudo-random stimuli sequence and the identities of CS1+ and CS2+ reversed in the other half of the participants. For both experiments, the stimulus colors were counterbalanced across participants[7,10,33,34].

The US (electric shocks) were delivered to the right wrist of participants via a DS-5 Isolated Bipolar Constant Current Stimulator (Digitimer, Welwyn Garden City, UK). The US levels were set by the participants themselves, starting from a slight level of shock (5 v) and gradually increase and settle to a level that they described as "uncomfortable, but not painful" (the maximum level was 10 V). The duration of all electric shocks was 200 ms with a current of 50 pluses per second.

SCRs were collected using two Ag-AgCI electrodes attached to the tips of the index and middle fingers of each subject's left hand. All skin conductance data were recorded via the Biopac® MP160 BioNomadix System, and analyzed using the Acknowledgement 5.0 software. The SCR level for each trial was calculated as the amplitude difference (in microsiemens) from peak to trough during the 0.5–4.5 s window after the colors stimuli (CS) onset. Responses below 0.02 µS were encoded as zero[5,7,10,31,33,34]. The raw SCR were divided by each subject's mean US responses and then square root transformed to normalize distributions.

The perception of the control ability over intrusions was measured by the 25-item TCAQ scale[18]. Participants were asked to rate on a five-point Likert-type scale the extent to which they agree with the statement from 1 (completely disagree) to 5 (completely agree). The internal consistency (Cronbach's alpha) of TCAQ scale was 0.92, and the reliability coefficient was satisfactory ($r = 0.88$).

**Experimental procedure**. Both experiments were carried out for 2 consecutive days with three stages. Before the experiments, all participants gave informed consent. During the experiments, subjects were required to stay relaxed and still, focus on the computer screen, and pay attention to the relationship between color stimuli (CS) and the shock (US).

Day1: Acquisition. During acquisition phases in two experiments, CS1+, CS2+, and CS– were presented ten times each for 4 s without the US, and intermixed with 12 trials where CS1+ (6 trials) and CS2+ (6 trials) co-terminated with the shock. Inter trial intervals (ITI) ranged from 10 to 12 s.

Day2: Extinction. During extinction training in Experiment 1 (suppression experiment), the CS1+, CS2+, and CS– were presented ten times each (lasting 4 s without shock). Before each CS1+ (active suppression) trial, a "No Think" instruction was displayed on the computer screen and lasted for 1 s to remind participants to actively avoid thinking about the potential consequence associated with the upcoming stimuli (CS1+). More specifically, participants were asked to blank their mind and keep fixation on the CS1+. If subjects accidentally thought about the CS1+-related outcome, they should immediately push the thoughts out of their consciousness. Participants were told to stop this process at the end of the trial. For all the CS2+ (suppression control) and CS– trials, an empty screen preceded the CS presentation instead and no such suppression was required.

During extinction phase in Experiment 2 (diversion experiment), participants were presented with the "Imagine" instruction for 1 s before CS1+ display and they were asked to imagine diversionary soothing natural scenes when the CS1+ (thought diversion) was presented on the computer screen. For example, if the square color of CS1+ is blue, subjects can come up with the diversionary thought of the blue sky. During the presentation of CS1+, participants were asked to imagine the natural scenes and keep fixation on the CS1+. Participants were told to stop the diversionary thought after each CS1+ trial ended. For all the CS2+ (diversion control) and CS– trials, an empty screen preceded the CS presentation instead and no such imagination was required.

Day2: Reinstatement and test phases. For both experiments, 30 min after extinction training, all participants received four un-signaled electric shocks with 10–12 s ITI (reinstatement). Eighteen seconds later, all subjects were presented with CS1+, CS2+, and CS– ten times each without electric shocks and their SCRs were recorded (test phase).

**Statistical and reproducibility**. To minimize the contamination of US-related SCRs on data analysis, we only included non-reinforced trials in the statistical analyses for both experiments (Fig. 2a, b).

To assess the recovery of fear response after reinstatement in both experiments, we conducted a mixed-model three-way ANOVA with the within-subjects factors CS+ (CS1+ vs. CS2+, defined by the mean SCR differences between CS+ and CS–), phase (last trial of extinction vs. first trial of test), and between-subjects factor experiment (suppression vs. diversion). We selected the last trial of extinction and the first trial of test phases due to the rapid adaptation of fear responses documented in the relevant literatures and it was a common practice to focus analysis on a limited number of trials[7,10,32,34–41].

There were significant effects of experiment ($F_{1,53} = 4.171$, $P = 0.046$, $\eta^2 = 0.073$) and experiment × phase interaction ($F_{1,53} = 14.57$, $P < 0.001$, $\eta^2 = 0.216$), suggesting active suppression and thought diversion have different effects on fear reinstatement. More specifically, there was no significant effect of phase ($F_{1,27} = 3.629$, $P = 0.067$, $\eta^2 = 0.118$), CS+ ($F_{1,27} = 0.482$, $P = 0.493$, $\eta^2 = 0.018$), or their interaction ($F_{1,27} = 0.104$, $P = 0.75$, $\eta^2 = 0.004$) in Experiment 1. Post-hoc $t$-tests showed that actively

suppressing fear retrieval in the extinction not only successfully eliminated fear reinstatement for the targeted CS (CS1+, $t_{27} = -1.844$, $P = 0.076$), but also for non-related CS (CS2+, $t_{27} = -1.333$, $P = 0.194$) in the test phase (Fig. 2c), indicating a cue-independent effect of active suppression.

Moreover, the differential SCR responses induced by CS1+ and CS2+ in the test phase (defined as the mean SCR differences between CS+ and CS−) were not significantly different in Experiment 1 ($t_{27} = -0.322$, $P = 0.75$; Fig. 2c). However, thought diversion in Experiment 2 yielded different results: reinstatement effects were significantly different between CS (phase × CS+ interaction: $F_{1,26} = 10.626$, $P = 0.003$, $\eta^2 = 0.29$). Indeed, diversionary thought targeting CS1+ only eliminated the fear reinstatement for CS1+ ($t_{26} = -0.992$, $P = 0.33$), leaving the fear SCR response to CS2+ intact ($t_{26} = -4.633$, $P < 0.001$) and there was a significant difference between CS1+ and CS2 ($t_{26} = -3.26$, $P = 0.003$; Fig. 2c), suggesting a cue-specific effect for thought diversion (Fig. 2c). To rule out the possibility that different CS− response baseline might play a role in the results we observed, we conducted a mixed-model three-way analysis of covariance (ANCOVA) with the within-subjects factors CS+ (CS1+ vs. CS2+, defined as the SCR differences between CS+ and CS−), phase (last trial of extinction vs. first trial of test), between-subjects factor experiment (suppression vs. diversion), and the covariate CS− (the difference between the first test trial and the last extinction trial for CS−). There were significant effects of experiment ($F_{1,52} = 5.118$, $P = 0.028$, $\eta^2 = 0.090$) and experiment × phase interaction ($F_{1,52} = 4.982$, $P = 0.030$, $\eta^2 = 0.087$), suggesting active suppression and thought diversion still have different effects on fear reinstatement after including the CS− response as a covariate. More specifically, a two-way repeated measure ANCOVA with the factors of CS+ (CS1+ vs. CS2+, defined as the SCR differences between CS+ and CS−), phase (last trial of extinction vs. first trial of test), and the covariate CS− showed no significant effect of phase ($F_{1,26} = 1.184$, $P = 0.287$, $\eta^2 = 0.044$), CS+ ($F_{1,26} = 0.709$, $P = 0.408$, $\eta^2 = 0.027$), or their interaction ($F_{1,26} = 0.023$, $P = 0.882$, $\eta^2 = 0.001$) in Experiment 1. Similar ANCOVA was also conducted on Experiment 2 and we found that reinstatement effects were significantly different between CS1+ and CS2+ (phase × CS+ interaction: $F_{1,25} = 10.025$, $P = 0.004$, $\eta^2 = 0.286$). Therefore, our results remain robust after controlling for the individual difference of the CS− level.

Finally, to examine whether the thought-control abilities affect the level of fear recovery, we first split participants in both experiments into high and low thought-control abilities according to their TCAQ scores (median split, high control: mean = 80.64, SD = 6.57, n = 14 and low control: mean = 62.14, SD = 8.81, n = 14 in Experiment 1; high control: mean = 80.93, SD = 8, n = 14 and low control: mean = 60.2, SD = 9.5, n = 13 in Experiment 2). Besides, the TCAQ scores of participants were not statistically different between two experiments ($t_{53} = 0.132$, $P = 0.895$). We then conducted a mixed-model three-way ANOVA with between-subject factors of experiment (suppression vs. diversion), thought-control ability (high vs. low control abilities), and within-subjects factor of CS+ (CS1+ vs. CS2+) on fear recovery index (defined as the difference between the first test trial and the last extinction trial for CS1+ and CS2+, corrected for the corresponding CS− SCR baselines).

In line with our hypothesis, this analysis showed significant interaction effects of experiment × CS+ ($F_{1,51} = 4.552$, $P = 0.038$, $\eta^2 = 0.082$) and experiment × control ability ($F_{1,51} = 6.744$, $P = 0.012$, $\eta^2 = 0.117$). Specifically, we found a significant effect of control ability ($F_{1,26} = 9.017$, $P = 0.006$, $\eta^2 = 0.258$), but no effects of CS+ ($F_{1,26} = 0.1$, $P = 0.754$, $\eta^2 = 0.004$) or their interaction ($F_{1,26} = 0.006$, $P = 0.939$, $\eta^2 = 0.001$) in Experiment 1. Post-hoc t-tests showed that the fear recovery index of high-control participants was lower than that of low-control participants (CS1+: $t_{26} = 2.439$, $P = 0.02$; CS2+: $t_{26} = 2.288$, $P = 0.02$; Fig. 3a). However, there was no effect of thought-control ability ($F_{1,25} = 0.736$, $P = 0.399$, $\eta^2 = 0.029$) or control ability × CS+ interaction ($F_{1,25} = 0.273$, $P = 0.606$, $\eta^2 = 0.011$), but only a significant effect of CS+ ($F_{1,25} = 10.439$, $P = 0.003$, $\eta^2 = 0.295$) in Experiment 2. Post-hoc t-tests showed that there was no difference between low- and high-control participants in the fear recovery index (CS1+: $t_{25} = -1.028$, $P = 0.328$; CS2+: $t_{25} = -0.391$, $P = 0.707$; Fig. 3b), but only a significant difference between CS+ (CS1+ vs. CS2+; $t_{26} = -3.26$, $P = 0.003$).

Importantly, there was no difference between high-control and low-control participants of both experiments in the late phase of acquisition and last trial of extinction (all $P$ values > 0.10), excluding the possibility of thought-control ability influencing fear acquisition or extinction directly. In summary, these results indicate that thought-control ability is only related to the diminishment of fear response induced by direct active suppression (but not thought diversion) during the test phase.

**Reporting summary**. Further information on research design is available in the Nature Research Reporting Summary linked to this article.

## Data availability

All data of this study are available for download at https://osf.io/qvt6r/. The source data underlying Fig. 2a are provided as Supplementary Data 1, the source data underlying Fig. 2b as Supplementary Data 2, the source data underlying Fig. 2c as Supplementary Data 3, and the source data underlying Fig. 3 as Supplementary Data 4.

## Code availability

Software used for data analysis in this study was IBM SPSS Statics 23. The SPSS syntax is available for download at https://osf.io/qvt6r/.

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

## Acknowledgements

This work was supported by the National Natural Science Foundation of China Grants Nos. 31871140 and 32071090 (to J.L.) and National Institute of Health grant MH105535 (to D.S.).

## Author contributions

Y.W., Z.Z., D.S., and J.L. conceived and designed study; Y.W. and J.H. analyzed data; J.L. supervised data collection; Y.W., Z.Z., J.H., D.S., and J.L. wrote the paper.

## Competing interests

The authors declare no competing interests.
