## [Peer Review File · Communications Biology]

Reviewers' comments:

Reviewer #1 (Remarks to the Author):

This study examines the effects of two active suppression procedures, thought suppression and thought diversion, on return of fear responses (skin conductance response) following reinstatement. The results show that fear response is reduced for both a trained CS+ and an untrained CS+ following the thought suppression procedure, while fear is selectively reduced for a trained CS+ but not for an untrained CS+ following the thought diversion procedure. The study is interesting, somewhat novel, and meaningful especially considering the potential clinical application of the low cost procedures introduced here. That said, I have some major concerns regarding the data exclusion criteria and transparency, as well as several minor comments.

1) Exclusion criteria

It is common to exclude participants who do not show any sign of fear acquisition, but the authors need to justify their further exclusion of 12 participants who did not show fear extinction learning effects. This is because, besides it being less common, the interest of this study is the success of fear extinction (through various thought-methods) as well as its individual variabilities. Excluding those who did not succeed (who contribute to important individual variabilities) sheds a question on the generalizability of the results. This exclusion is even more puzzling when looking at the results of included participants. Specifically, the authors state their exclusion criteria as "twelve participants (n = 8 & 4, respectively) failed to show the evidence of fear extinction on day 2, since (a) their CS+ responses were larger than the CS- responses in both the last trial and (b) the latter half trials of extinction and (c) the difference between the CS+ response and the CS- response increased during fear extinction." First, please clarify if the excluded participants met all three criteria (a) to (c), and please clarify how you quantitatively defined (c). Second, the main results of included participants in Figure 2C (middle panel) show substantial proportions of participants who do show negative value of differential SCR, meaning their SCR to CS+ was lower than CS- on the last trial of extinction, which means that some of those included participants at least meet the exclusion criterion (a). Could the authors clarify and demonstrate that included participants all showed positive value of differential SCR in the latter half trials of extinction (i.e., *not* meeting criterion (b)) (an additional figure is appreciated) and that those included participants also do not meet criterion (c). Third, could the authors provide a supplemental figure showing that inclusion of those excluded 12 participants does not (at least qualitatively) change the main results?

2) Manipulation check

Did the authors check the trial-wise and/or session-wise success of participants' thought control during the experiments? In other words, how successful were they to follow the instructions to actively suppress their thoughts in some trials (with CS+1) but not in other trials (with CS+2)? Were their successes equivalent between Experiment 1 and 2? If these were not directly measured, although less preferable, could the authors provide major papers to clarify that not using manipulation check is the common practice in the field.

3) Rationale for thought diversion

Relative to the active thought procedure, rationale and details (including its background and previous studies) on thought diversion is missing in the introduction, making it difficult to evaluate the authors' hypothesis and results. Where does their hypothesis regarding thought diversion come from?

4) Power analysis

Could the authors justify the use of averaged effect size across multiple studies when I assume that those studies used non-identical statistical tests and sample sizes? Would it be more appropriate to estimate sample size based on each of the studies then average the required N (or take the most conservative (i.e., largest) N)? If you can justify the averaging of effect size, please provide the rationale, preferably with references.

After the authors state their planned N of 23 for each Experiment, they suddenly state they collected N of 90, then list their exclusion methods. Please clarify that the authors collected the data until they collected the participants meeting inclusion criteria, if this was the case.

5) Overall clarity

I understand that writing with a tight word limit is difficult. However, there are numbers of paragraphs and sections that are unclear and can be improved with re-writing and re-organization.

Related to the point 3) mentioned above, descriptions of thought diversion comes later than the authors' introduction of their study aim with thought diversion in line 70. The paper could benefit from some reorganization to introduce some important details first. Some statistics in the main text are hard to interpret without Online Methods. For example, it is unclear what is the measurement in the ANOVA (line 105), is this differential SCR or raw SCR? Either case, how does the significant main effect of trial (rather than the contrast between CS+ versus CS-, or contrast between differential SCR to CS+ and zero) justify the establishment of fear acquisition?

Also, important non-significant effects (such as absence of significant main effect of the factor, Experiment, and interactions involving Experiment) need to be mentioned. Similarly, it is difficult to comprehend how the absence of significant effect implies the successful fear reduction unless the authors clarify what is being contrasted in subsequent t-tests (line 123). The term such as "phase" in the ANOVAs need to be briefly defined in the main text.

It is difficult to understand whether the study was within or between subject design unless the reader reaches Online Methods.

While there are some important details missing as above, some points in the introduction and discussion (e.g., on reconsolidation) seem not directly related to this study and can be omitted to save space.

Besides these specific examples, there are some other minor sections that can be improved, e.g., defining and consistent usage of terminologies.

Reviewer #3 (Remarks to the Author):

REVIEW COMMSBIO-20-2649-T

Active suppression prevents the return of threat memory in humans

AUTHORS: Y. Wang, Z. Zhu, J. Hu, D. Schiller, J. Li

SUMMARY:

The manuscript examines the hypothesis that suppression can also eliminate a fear response. Specifically, it tests the hypothesis that thought suppression impairs the fear memory itself irrespective of the retrieval cue. By contrast, thought diversion should only impair the fear response associated with the unique retrieval cue. Finally, the authors examined the impact of individual differences in thought control ability on these effects. To test these hypotheses, they conducted two experiments combining the Think/No-Think (TNT) procedure with a double-cue conditioning procedure. In both experiments, two neutral CS+ were paired with an electric shock, resulting in an increased skin conductance response towards those stimuli. Experiments then only differed in the extinction phase: in experiment 1, participants actively suppressed memories of the aversive outcome associated with one CS+. In experiment 2, participants recalled an alternative pleasant thought when confronted with the CS+. The return of fear was diminished for both CS+ following suppression, whereas it was only diminished for the targeted CS+ following diversion. Moreover, fear recovery was lower for the high control group, but only in the suppression experiment.

OVERALL COMMENTS:

The authors examine a critical question regarding the impact of suppression on affective responses of a fear memory. This work has thus potentially important translational implications. As such, we think that it will be of interest to a wider audience. The procedure is well designed and the results nicely support the authors' hypotheses. However, we do think that the manuscript could further be strengthened by some conceptual and methodological clarifications, and a more careful discussion. We will elaborate on each of those points below (in no particular order).

DETAILED COMMENTS:

(1) At times, we found the statistical analysis difficult to follow that had been reported in the manuscript proper. This is probably a consequence of the likely word limit of the journal. However, as it stands, the description of, e.g., the ANOVAs are unclear. For example, it is not clear whether the analyses of the acquisition and extinction phase entails data from the CS-, or what is meant by experiment x phase during reinstatement. The analyses – and corresponding results- are reported much more detailed in the methods. We suggest to shift some of these to the result section. (Btw., it is also unclear what supplement the authors refer to. We took this to mean said online methods.)

(2) The authors differentiate thought suppression and thought diversion. There is of course nothing wrong with adopting this terminology. However, these strategies are more commonly referred to as direct suppression and thought substitution. In fact, thought suppression is probably more related to the distinct work by Daniel Wegner et al.

(3) We have trouble following the power analysis. Obviously, the experiment was sufficiently powered given that it did yield a significant effect. However, the authors report to base this on a main effect for reinstatement yet perform the analysis for a three-way interaction (including a between-subject factor.) We thus wonder whether the analysis is really valid. We also don't quite understand how the outcome of the analysis then matches the number of recruited participants?

(4) What was the rationale for the 24 h gap between acquisition and the remainder of the procedure? Indeed, the fact that suppression targets consolidated memories is particularly noteworthy. This should probably be emphasized and discussed in more detail.

(5) The authors argue that suppression eliminates the fear response and interpret the absence of fear reinstatement as the consequence of this. However, one could also argue that the extinction procedure itself (independent of the suppression instructions) yielded the diminished fear response. As there is no difference between CS1+ and CS2+ in the extinction or reinstatement phase, one cannot really disentangle the effects of suppression and extinction. The authors should thus stress more clearly that this interpretation hinges on the interaction across experiments.

(6) In the test phase, only the difference between Cs+ and Cs- enters the analyses. However, Fig. 2 suggests that the difference in the CS- response could drive that effect, given that the initial responses towards the CS- seem to be much higher in experiment 1 than in experiment 2. Could that be ruled out?

(7) Almost 40% of participants were excluded. The exclusion criteria are transparently reported, still such a high exclusion rate certainly seems to limit the generalizability of the results to the whole population. We wonder whether the sample might be biased towards anxious participants that are also very good suppressors. (i) Did the excluded participants differ in terms of control ability and (ii) how do the results look like with the full sample (or at least including the "non-learners")?

(8) To foster any attempts to replicate this important result pattern, the manuscript could provide more details regarding the following methodological points:

(a) How exactly was the pseudo-random presentation order created, both for the extinction and test

phase? This seems important, given the big influence of first and last stimulus and sequence.

(b) How was the thresholding procedure implemented? Was there a standardized procedure and how many US did it involve, given that the number of US given during this procedure might already have an impact in habituation. What was the mean intensity (+ variance)?

(c) What was on the screen during the ITI?

(d) Where the measuring electrodes attached to the Non-dominant hand? Was any NaCl electrode paste used?

(9) The authors focused their analysis on a very limited number of trials (e.g., late phase acquisition vs. last trial of extinction; last trial extinction vs. first trials reinstatement). We understand that these effects are quite sensitive and, by their very nature, change over time. However, we would suggest that the manuscripts should provide a rationale for these choices and indicates whether they are common and accepted in the pertinent literature.

(10) A precise description is also important in the case of TNT procedures. The authors could clarify on the instructions. What were the verbatim instructions? Where participants asked to suppress the sensation or their knowledge about the aversive history associated with a CS? Was there any practice?

(11) It is great that data that is available on OSF. However, it would be helpful for the potential reader if

(a) the data were published in a neutral format (e.g., .csv) that does not require specific proprietary software.

(b) they were accompanied by a simple read_me file that explains data format, variables and abbreviations.

(c) We assume analyses were done with SPSS (not mentioned in the manuscript). However to reproduce the analyses, the authors may consider also uploading the SPSS syntax.

Minor details.

(12) line 122: For readers not familiar with conditioning, a brief explanation of CS1 and CS2 might be helpful.

(13) When discussing evidence for suppression-induced forgetting and its possible impairment in people with low control ability, the authors may wish to discuss the recent meta-analysis by Stramaccia et al. and the recent article by Mary et al. about the Paris terrorist attack.

Responses to Reviews

We thank our editor and the reviewers for their detailed comments and constructive suggestions. We have incorporated their feedbacks in the revised manuscript and responded to each of the comments below. We strongly believe that the changes we made in light of the comments have significantly improved our submission (COMMSBIO-20-2649A). We are also happy to address any additional comments, or to clarify our responses if we did not address a concern to the reviewer's satisfaction. Below, we provide a point-by-point response in which we delineate the points raised by the editor and reviewers (in *italics*), followed by our response and a description of how each point was addressed in the revised manuscript (**red**).

COMMENTS FROM THE REVIEWERS:

Reviewer #1

1. *“The study is interesting, somewhat novel, and meaningful especially considering the potential clinical application of the low cost procedures introduced here... but the authors need to justify their further exclusion of 12 participants who did not show fear extinction learning effects. ... Excluding those who did not succeed (who contribute to important individual variabilities) sheds a question on the generalizability of the results... First, please clarify if the excluded participants met all three criteria (a) to (c), and please clarify how you quantitatively defined (c)...”*

We truly appreciate the positive comments from the reviewer and agree with the reviewer that cautions should be taken when excluding subjects in the extinction training. We only excluded subjects who met all the criteria from (a) to (c) during the extinction training. And for the criterion (c), we focused on the difference between CS+

and the CS- responses, and the exclusion criterion (c) was defined as the difference of the last trial of extinction was numerically larger than the difference of the first trial of extinction, AND the mean difference of latter half trials was larger than that of the first half trials in the extinction, AND the mean difference of latter half trials was larger than the first trial difference in the extinction, AND the last trial difference was larger than the mean difference of the first half trials in the extinction. Therefore, we excluded 12 participants in two experiments following such stringent criteria, which was also adopted by several recent studies (Schiller et al.,2010; Schiller, Kanen, LeDoux, Monfils, & Phelps, 2013; Raio, Hartley, Orederu, Li, & Phelps, 2017). We have added the details of exclusion criterion (c) into the main text:

“The increase of SCR difference (CS+ vs. CS-) during extinction was defined as the difference of the last trial larger than that of the first trial of extinction, and the mean difference of latter half trials larger than that of the first half trials in the extinction, and the mean difference of latter half trials larger than that of the first trial, and the last trial difference larger than the mean difference of the first half trials in the extinction^{7,31,32,}”

2. “...Second, the main results of included participants in Figure 2C (middle panel) show substantial proportions of participants who do show negative value of differential SCR, meaning their SCR to CS+ was lower than CS- on the last trial of extinction, which means that some of those included participants at least meet the exclusion criterion (a). Could the authors clarify and demonstrate that included participants all showed positive value of differential SCR in the latter half trials of extinction (i.e., *not* meeting criterion (b)) (an additional figure is appreciated) and that those included participants also do not meet criterion (c)...”

The reviewer is right that some of the participants included in the analysis did meet criterion (a) (CS+ response was larger than the CS- response in the last trial of extinction). However, as we mentioned above, we only excluded participants who met all of the exclusion criteria from (a) to (c). Therefore, some of the included subjects met exclusion criterion (a) (n = 17 & 10 for experiments 1 & 2, respectively); some met

criterion (b) (n = 25 & 19, respectively) and some met criterion (c) (n = 8 & 4, respectively). In experiment 1, we first identified 17 participants who met criterion (a), and subsequently identified 16 (out of the 17) subjects who met criterion (b), 8 of which (out of 16) also met criterion (c). Therefore, we excluded 8 participants in experiment 1. Similarly, 10 subjects met criteria (a) & (b), out of which 4 subjects also met criterion (c) and we excluded 4 participants in experiment 2.

3. *“...Third, could the authors provide a supplemental figure showing that inclusion of those excluded 12 participants does not (at least qualitatively) change the main results?...”*

Following the reviewer’s suggestion, we added the excluded 12 participants into the original 55 participants (a total of 67 subjects) and performed the analysis again (n = 36 & 31 for experiment 1 & 2, respectively). To better compare with the original results, we generated figures with similar layouts as following:

It is clear from the above figure that including those 12 participants does not change the main results (compared to the original figure 2). Specifically, the 3-way interaction among CS+ (CS1+ vs. CS2+), phase (1st trial of test vs. last trial of extinction) and experiment (exp1 vs. exp2) is still significant ($F_{1,65} = 6.405$, $P = 0.014$, $\eta^2 = 0.090$, panel

a & b in the figure above) and such interaction is mainly driven by the difference of the phase x CS+ interaction in experiment 2 ($F_{1,30} = 16.279$, $P < 0.001$, $\eta^2 = 0.352$, panel c).

Importantly, we also performed the analysis of fear recovery index as a function of thought control abilities. As we can see from the following figure, adding 12 subjects back to the analysis produced almost identical results (compared to original figure 3): higher thought control ability participants had less fear recovery in experiment 1 ($F_{1,34} = 11.218$, $P = 0.002$, $\eta^2 = 0.248$) and there was no difference between CS1+ and CS2+ ($F_{1,34} = 0.129$, $P = 0.721$, $\eta^2 = 0.004$) nor CS+ x control ability interaction ($F_{1,34} = 0.007$, $P = 0.932$, $\eta^2 < 0.001$). However, in experiment 2, CS+ had a significant main effect ($F_{1,29} = 15.691$, $P < 0.001$, $\eta^2 = 0.351$), but thought control ability ($F_{1,29} = 0.319$, $P = 0.577$, $\eta^2 = 0.011$) and CS+ x control ability ($F_{1,29} = 0.036$, $P = 0.851$, $\eta^2 = 0.001$) had no effects on fear recovery.

4. "...Did the authors check the trial-wise and/or session-wise success of participants' thought control during the experiments? In other words, how successful were they to follow the instructions to actively suppress their thoughts in some trials (with CS+1) but not in other trials (with CS+2)? Were their successes equivalent between Experiment 1 and 2? If these were not directly measured, although less preferable, could the authors provide major

papers to clarify that not using manipulation check is the common practice in the field.”

We did not check the trial-wise success of participants' thought control during the experiment for the fear that doing so might prompt subjects to recall the exact thoughts that they were supposed to inhibit or divert. In fact, not using manipulation check during the experiment is the common practice in the thought control literatures (Anderson & Green, 2001; Anderson, 2004; Johnson & Anderson, 2004; Bergström, de Fockert, & Richardson-Klavehn, 2009; Anderson, Reinholz, Kuhl, & Mayr, 2011; Racsmány, Conway, Keresztes, & Krajcsi, 2012; Benoit & Anderson, 2012; Paz-Alonso, Bunge, Anderson, & Ghetti, 2013; Weller, Anderson, Gómez-Ariza, & Bajo, 2013; Gagnepain, Henson, & Anderson, 2014; Catarino, Küpper, Werner-Seidler, Dalgleish, & Anderson, 2015; Wang, Cao, Zhu, Cai, & Wu, 2015; Hulbert & Anderson, 2018).

5. *“... rational and details (including its background and previous studies) on thought diversion is missing in the introduction, making it difficult to evaluate the authors' hypothesis and results. Where does their hypothesis regarding thought diversion come from?”*

We are sorry for not providing clear introduction about thought diversion and have since added the following sentences into the introduction:

“Other mnemonic control methods such as thought substitution and thought diversion, are based on interference theory and suggest that specific target memory can be interrupted during recall if there are multiple cue-associated memories competing for the cue-triggering memory retrieval. For example, in memory interference studies, established cue-episodic memory recall was reduced when the participants were asked to associate the cue with a substitute memory target^{13,14,15,16}. However, compared to direct memory suppression, memory interference effect is believed to be cue-specific and cannot generalize to other target memory associated cues^{13,15,16}.”

6. *“Could the authors justify the use of averaged effect size across multiple studies when I assume that those studies used non-identical statistical tests and sample sizes? Would it be more appropriate to estimate sample size based on each of the studies then average the required N (or take the most conservative (i.e., largest) N)? If you can justify the averaging of effect size, please provide the rationale, preferably with references. After the authors state their planned N of 23 for each Experiment, they suddenly state they collected N of 90, then list their exclusion methods. Please clarify that the authors collected the data until they collected the participants meeting inclusion criteria, if this was the case.”*

We are sorry about the sudden transition in the methods section when describing the sample size. In fact, the other reviewer also pointed out this issue. Before we started to collect the data sample, we set a preliminary target of 90 participants knowing in advance that a rather high percentage of subjects might be excluded due to the well-documented SCR phenomena of non-respondents and non-learners (Schiller et al.,2010; Schiller et al.,2013; Raio et al.,2017). After finishing the data collection, we applied the data exclusion criteria and eventually ended up with 55 final participants which were above the threshold of 48 and we believed the sample size was enough for our proposed experiment data analysis. The fact that our results remained robust even after we added the 12 originally excluded subjects back into the data analysis (see our response to comment #3) further confirmed that the effects we observed were reliable.

Since there were no previous studies directly comparing the effect of active suppression and thought diversion in fear learning, our sample size estimation was based on studies that compared different reconsolidation methods on fear return. The studies we referenced used different sample sizes (N = 40, 30, 56 total sample sizes, respectively) (Kindt, Soeter, & Vervliet, 2009; Kindt & Soeter, 2018; Thompson & Lipp, 2017). Since we estimated the sample size based on the partial η^2 ($\eta^2 = 0.19, 0.24, 0.03$) in the above referenced studies, we calculated the subject number according to the median η^2 (instead of the mislabeled mean η^2 in the original text). We also took the reviewer’s suggestion and calculated the required participant number first from each study, and such calculation yielded required participant number to be N = 48, 40, and

264, which suggest that the mean and median sample sizes should be 118 and 48, respectively. Our final sample size was 55, which we believe should give us enough power to detect the effect we set out to test. We have rewritten the methods section as the following:

“Due to the well-known high attrition rate of SCR data collection^{7,31,32} in recent fear learning experiments, we first recruited a total of 90 human subjects (44 females; mean age = 22.2, SD = 2.59).”

7. “... descriptions of thought diversion comes later than the authors' introduction of their study aim with thought diversion in line 70. The paper could benefit from some reorganization to introduce some important details first...”

We thank the reviewer's suggestion and has added descriptions of thought diversion earlier in the introduction to better present the idea of both memory suppression (active suppression) and interference (thought diversion).

8. “...For example, it is unclear what is the measurement in the ANOVA (line 105), is this differential SCR or raw SCR? Either case, how does the significant main effect of trial (rather than the contrast between CS+ versus CS-, or contrast between differential SCR to CS+ and zero) justify the establishment of fear acquisition?”

We are sorry for the lack of clarity in the description of results. The measurement, as the reviewer correctly pointed out, is the differential SCR of both CS+ and CS- (the difference of CS1+ and CS-, CS2+ and CS-, respectively). We did not look at the contrast between CS+ versus CS-, since such a contrast might reflect an overall difference between CS+ and CS-, and it is not necessarily associated with learning (i.e., trial independent). Instead, a main effect of trial on differential SCR is effectively a test of the interaction effect between CS+/CS- and trials. That being said, we agree with the reviewer that our approach can be improved by directly testing the interaction effect of CS+/CS- and trial during fear acquisition and extinction. We therefore performed extra analyses: Fear acquisition and extinction were assessed using mixed-model 2

(experiment) \times 3 (CS1+, CS2+ and CS-) \times 10 (trial) three-way ANOVAs. In acquisition, the mixed-model 3-way ANOVA showed significant main effects of trial (trial; $F_{9,477} = 6.888$, $P < 0.001$, $\eta^2 = 0.115$) and CS (CS1+, CS2+ and CS-; $F_{2,106} = 116.32$, $P < 0.001$, $\eta^2 = 0.687$). Meanwhile, the interaction effect of trial and CS was significant ($F_{18,954} = 2.936$, $P < 0.001$, $\eta^2 = 0.052$), but the no effect of experiment (suppression vs. diversion), experiment \times CS, experiment \times trial, or experiment \times CS \times trial (all p values > 0.1).

In extinction, the mixed-model 3-way ANOVA showed significant main effects of trial (trial; $F_{9,477} = 32.263$, $P < 0.001$, $\eta^2 = 0.378$) and CS (CS1+, CS2+ and CS-; $F_{2,106} = 21.240$, $P < 0.001$, $\eta^2 = 0.286$). Meanwhile, interactions of trial \times CS interaction and CS \times experiment were significant ($F_{18,954} = 5.339$, $P < 0.001$, $\eta^2 = 0.092$; $F_{2,106} = 3.469$, $P = 0.035$, $\eta^2 = 0.061$), but the no effect of experiment (suppression vs. diversion), experiment \times CS, experiment \times trial, or experiment \times CS \times trial (all p values > 0.1).

Since the CS \times trial interaction effects are significant in the fear acquisition and extinction phases in both experiments, we have confirmed that learning and extinction did occur in our experiments and made clear in the text that the differential SCR was used as dependent variable.

9. *“Also, important non-significant effects (such as absence of significant main effect of the factor, Experiment, and interactions involving Experiment) need to be mentioned. Similarly, it is difficult to comprehend how the absence of significant effect implies the successful fear reduction unless the authors clarify what is being contrasted in subsequent t-tests (line 123)... “*

The t-tests on line 123 demonstrated that the reinstatement effect was absent for both CS1+ and CS2+ during the test phase (the SCRs induced by the CS1+ and CS2+ were not different from those of CS-), indicating successful inhibition of the return of fear responses during test. We have rewritten the results section to make the results more comprehensive and included important non-significant effects in the text.

10. *“... The term such as “phase” in the ANOVAs need to be briefly defined in the main text”*

We have defined the term “phase” in the main text and it reads “Both experiments can be designated into three phases: acquisition, extinction and test (Fig. 1).”

11. *“...there are some other minor sections that can be improved, e.g., defining and consistent usage of terminologies.”*

We thank the reviewer for the thorough reading of our manuscript and we have made corresponding changes in the main text accordingly. For example, we have adopted the term active suppression throughout the manuscript to refer to experiment 1.

Reviewer #3

1. *“This work has thus potentially important translational implications. As such, we think that it will be of interest to a wider audience. The procedure is well designed and the results nicely support the authors’ hypotheses. However, we do think that the manuscript could further be strengthened by some conceptual and methodological clarifications, and a more careful discussion.”*

We appreciate the positive comments from the reviewer and have since significantly revised the manuscript.

2. *“...This is probably a consequence of the likely word limit of the journal. However, as it stands, the description of, e.g., the ANOVAs are unclear. For example, it is not clear whether the analyses of the acquisition and extinction phase entails data from the CS-, or what is meant by experiment x phase during reinstatement. The analyses – and corresponding results- are reported much more detailed in the methods. We suggest to shift some of these to the result section.”*

We are sorry for not presenting the results clearer. In fact, reviewer 1 expressed similar concerns regarding the term “phase” and the dependent variables used in the ANOVA analyses. We have combed through the result section to make sure the descriptions are more comprehensive.

3. *“... it is also unclear what supplement the authors refer to. We took this to mean said online methods”*

The reviewer is correct that terms should have been used consistently throughout the manuscript. In keeping with the format of Communications Biology, we have changed “Online Methods” to Supplementary methods.

4. *“The authors differentiate thought suppression and thought diversion. There is of course nothing wrong with adopting this terminology. However, these strategies are more commonly referred to as direct suppression and thought substitution...”*

We thank the keen observation of the reviewer. We originally adopted the terms “thought suppression” and “thought diversion” to differentiate the goals of two experiments. As the reviewer correctly pointed out, “thought suppression” is more related to the well-accepted “white bear” effect and we have since changed the term to active suppression to avoid confusion. As for thought substitution, it is typically defined as “a memory control process that involves avoiding recall of a memory when confronted with a reminder by using the reminder to recall an alternative memory” (Engen & Anderson 2018). According to the previous literature, the effects of thought substitution could be mixed: some studies showed that thought substitution induced amnesia in both independent-cue and dependent-cue tests, suggesting a generalized memory deficit (Benoit & Anderson 2012; Noreen & Ridout 2016). However, other studies using thought substitution method identified a cue-dependent amnesia effect (Bergström et al., 2009; Racsmány et al., 2012). The discrepancy among these studies might be reconciled by the fact that traditional thought substitution emphasizes both “avoiding recall of the original memory” and “recall an alternative memory”, which correspond to active suppression and thought diversion, respectively. In fact, recent thought diversion literature that only asked participants to recall a new target memory tended to have cue-specific memory deficit (Wang et al., 2015). In our study, we adopted this approach and only instructed our participants to associate the CS+ with certain neutral target memory and did not mention that they needed to suppress the

learned consequence of the CS (US delivery). Therefore, the term “thought diversion” seems to be more appropriate in the context of our experiments.

5. *“...However, the authors report to base this on a main effect for reinstatement yet perform the analysis for a three-way interaction (including a between-subject factor.) We thus wonder whether the analysis is really valid.”*

We are sorry for the lack of clarity in the description of power estimation analysis. We set out to test whether there was significant difference between the reinstatement effects induced by the CS+ across both experiments since we hypothesized that active suppression would eliminate the fear reinstatement effects irrespective of the eliciting CS+ while thought diversion’s effect was CS+ specific. Therefore, we focused on the CS+ related SCR differences (relative to CS-) across two experiments, that is, the 3-way interaction. More specifically, we estimated the sample size based on the effect size of the three-way interaction (within-subject factor CS+ (CS1+ vs. CS2+), within-subject factor phase (extinction vs. test), and the between-subjects factor experiment (active suppression vs. thought diversion)) from previous literatures (Kindt et al.,2009; Kindt & Soeter, 2018; Thompson & Lipp, 2017), which had N = 40, 30, 56 total sample sizes, respectively). We started to collect data sample and set a preliminary target of 90 participants knowing in advance that a rather high percentage of subjects might be excluded due to the well-documented high attrition rate of SCR experiments (i.e., non-respondents and non-learners, Schiller et al.,2010; Schiller et al.,2013; Raio et al.,2017). Eventually, we ended up with 28 and 27 final participants in experiments 1 and 2, respectively.

6. *“What was the rationale for the 24 h gap between acquisition and the remainder of the procedure? Indeed, the fact that suppression targets consolidated memories is particularly noteworthy. This should probably be emphasized and discussed in more detail.”*

The reviewer’s point is well taken. Indeed, we meant to test the effectiveness of active suppression and thought diversion on consolidated associative memory. To emphasize this point, we have added the following sentence into the discussion:

“The issue is further complicated by the fact that in declarative memory research, mnemonic control methods were typically tested on memories that had not gone through consolidation process^{11,13,20} (usually less than one hour after the training of cue-target memory association in the laboratory experiments). However, fear related associative memory poses a greater challenge since it is typically well-established before the memory modulation attempt such as in posttraumatic stress disorder (PTSD) or phobia patients^{6,7,9,21,22}. Therefore, it remains to be tested whether declarative memory inhibition (active suppression) and interference (thought diversion) methods can be applied successfully to the consolidated fear memory.”

7. *“The authors argue that suppression eliminates the fear response and interpret the absence of fear reinstatement as the consequence of this. However, one could also argue that the extinction procedure itself (independent of the suppression instructions) yielded the diminished fear response. As there is no difference between CS1+ and CS2+ in the extinction or reinstatement phase, one cannot really disentangle the effects of suppression and extinction. The authors should thus stress more clearly that this interpretation hinges on the interaction across experiments. “*

Although the fear reinstatement effect is well-established, it is true that in experiment 1 we cannot theoretically eliminate the possibility that extinction itself diminished fear response during test phase. Therefore, the results in experiment 2 are critical to demonstrate that it was not extinction that drove the diminished fear response; rather, it should be active suppression (experiment 1) or thought diversion (experiment 2). We have clarified this point in the discussion:

“It is worth noting that both active suppression and thought diversion were applied to the consolidated fear memory which was established 24hr before the extinction training. Also, although fear reinstatement is a robust phenomenon in fear learning literature, our interpretation that active memory suppression eliminated fear response hinged on the results in experiment 2, where thought diversion was only effective on the CS+ it

specifically applied to. The other CS+, however, showed the stereotypical fear reinstatement effect.”

8. *“In the test phase, only the difference between Cs+ and Cs- enters the analyses. However, Fig. 2 suggests that the difference in the CS- response could drive that effect, given that the initial responses towards the CS- seem to be much higher in experiment 1 than in experiment 2. Could that be ruled out? “*

Given that we were interested in the difference between CS1+ and CS2+ after the correction from CS- baseline across experiments, different CS- response baseline should not play a role in the results we observed. However, to rule out this possibility in a principled approach, we conducted a mixed-model 3-way analysis of covariance (ANCOVA) with the within-subjects factors CS+ (CS1+ vs. CS2+, defined as the SCR differences between CS+ and CS-), phase (last trial of extinction vs. first trial of test), between-subjects factor experiment (suppression vs. diversion), and the covariate CS- (the difference between the first test trial and the last extinction trial for CS-). There were significant effects of experiment ($F_{1,52} = 5.118$, $P = 0.028$, $\eta^2 = 0.090$) and experiment \times phase interaction ($F_{1,52} = 4.982$, $P = 0.030$, $\eta^2 = 0.087$), suggesting active suppression and thought diversion still have different effects on fear reinstatement after including the CS- response as a covariate. For experiment 1, we conducted a 2-way repeated measure ANCOVA with the factors of CS+ (CS1+ vs. CS2+, defined as the SCR differences between CS+ and CS-), phase (last trial of extinction vs. first trial of test) and the covariate CS-. Consistent with our original results, there was no significant effect of phase ($F_{1,26} = 1.184$, $P = 0.287$, $\eta^2 = 0.044$), CS+ ($F_{1,26} = 0.709$, $P = 0.408$, $\eta^2 = 0.027$), or their interaction ($F_{1,26} = 0.023$, $P = 0.882$, $\eta^2 = 0.001$) in experiment 1. Similar ANCOVA was also conducted on experiment 2 and again, we found that reinstatement effects were significantly different between CS1+ and CS2+ (phase \times CS+ interaction: $F_{1,25} = 10.025$, $P = 0.004$, $\eta^2 = 0.286$). Therefore, our results remain robust after controlling for the individual difference of the CS- level.

9. “(i) Did the excluded participants differ in terms of control ability and (ii) how do the results look like with the full sample (or at least including the “non-learners”)?

Out of the 90 participants, 19 (8 females, mean age = 21.6, SD = 2.83) of them were “non-responders” (SCR responses consistently lower than 0.02 μ S for all the CS) and they quitted the experiments after the fear acquisition phase. Therefore, we did not have thought control ability data for these 19 subjects. Out of the remaining 71 participants, 4 subjects failed to show learning in the fear acquisition phase (n = 3 & 1 for experiment 1 & 2, respectively) and 12 participants did not show evidence of fear extinction (n = 8 & 4, respectively). So, in total, there were 16 “non-learners” (n = 11 & 5 for experiment 1 & 2, accordingly) in our experiments and they, unlike the “non-responders”, did finish the corresponding experiment and fulfil the control ability questionnaire. Although there is no difference in terms of thought control ability between the 11 “non-learners” and the final 28 subjects in experiment 1 (Mann-Whitney $U_{28,11} = 95$, $P = 0.067$) and no difference between the 5 “non-learners” and the 27 participants in experiment 2 (Mann-Whitney $U_{27,5} = 49.5$, $P = 0.361$). Importantly, similar to our response to Q3 by reviewer 1, which added 12 “non-extinctioners” into the data analysis, adding the 16 “non-learners” also yielded almost identical results:

It is clear from the above figure that including those 16 non-learners does not change the main results (compared to the original figure 2). Specifically, the 3-way interaction among CS+ (CS1+ vs. CS2+), phase (1st trial of test vs. last trial of extinction) and experiment (exp1 vs. exp2) is still significant ($F_{1,69} = 6.082$, $P = 0.016$, $\eta^2 = 0.081$, panel a & b in the figure above) and such interaction is mainly driven by the difference of the phase x CS+ interaction in experiment 2 ($F_{1,31} = 16.993$, $P < 0.001$, $\eta^2 = 0.354$, panel c).

Importantly, we also performed the analysis of fear recovery index as a function of thought control abilities. As we can see from the following figure, adding 16 non-learners back to the analysis produced almost identical results (compared to original figure 3): higher thought control ability participants had less fear recovery in experiment 1 ($F_{1,37} = 13.775$, $P = 0.001$, $\eta^2 = 0.271$) and there was no difference between CS1+ and CS2+ ($F_{1,37} = 0.203$, $P = 0.655$, $\eta^2 = 0.005$) nor CS+ x control ability interaction ($F_{1,37} = 0.221$, $P = 0.641$, $\eta^2 = 0.006$). However, in experiment 2, CS+ had a significant main effect ($F_{1,30} = 16.474$, $P < 0.001$, $\eta^2 = 0.354$), but thought control ability ($F_{1,30} = 0.405$, $P = 0.529$, $\eta^2 = 0.013$) and CS+ x control ability ($F_{1,30} = 0.053$, $P = 0.819$, $\eta^2 = 0.002$) had no effects on fear recovery.

We therefore conclude that our results are robust and not susceptible to the “non-learner” exclusion criteria and added the following sentence into the supplementary information:

“It should be noted that our main results remain robust after including the 4 non-learners and 12 non-extinctioners into the statistical analyses.”

10. *“How exactly was the pseudo-random presentation order created, both for the extinction and test phase?”*

In the extinction phase, each CS1+, CS2+ and CS- were presented 10 times without electric shocks. We generated a pseudo-random stimulus order for both experiments with the rule that no same trial-type (CS1+, CS2+, CS-) repeated more than twice and applied the same sequence to all the participants (the exact sequence has been uploaded with the data file). The square colors (red, yellow, and blue) for different CS (CS1+, CS2+ and CS-) were counterbalanced across subjects and stayed the same for each participant during the acquisition, extinction and test phases. Trial sequence for test phase was generated similar to that of the extinction phase with one major difference: To exclude the possibility that the potential SCR difference between CS1+ and CS2+ was simply caused by the presentation order of CS1+ and CS2+, we swapped the identities of CS1+ and CS2+ and generated a new sequence. During the test phase, participants were randomly assigned to either the un-swapped or swapped sequence to counterbalance the potential order effect caused by presenting CS1+ or CS2+ first.

11. *“How was the thresholding procedure implemented? Was there a standardized procedure and how many US did it involve, given that the number of US given during this procedure might already have an impact in habituation. What was the mean intensity (+ variance)?”*

Before the experiment on day 1, we calibrated electric shock threshold level for each participant. We informed the participants that they might experience a series of electric shocks to determine the final shock level used in the experiments. The shock level started from a rather low voltage (5v) and gradually increased with increment of 1v

until the participants indicated that the electric shock was ‘uncomfortable, but not painful’ (the maximum level was 10V). For experiment 1 the mean intensity is $5.57v \pm 0.78$, and for experiment 2 the mean intensity is $5.85v \pm 1.09$. Therefore, for most participants, they experienced 1-3 shocks before the actual experiment.

12. *“What was on the screen during the ITI?”*

There is a 10-12 seconds inter-trial-interval (ITI), during which a black screen was presented with a white fixation cross at the center of the computer screen.

13. *“Where the measuring electrodes attached to the Non-dominant hand? Was any NaCl electrode paste used?”*

We attached the measuring electrode to the index and middle fingers (between the first and second phalanges) of the Non-dominant hand and we used GEL 101 electrode paste (Biopac), which is an isotonic NaCl electrode paste for electrodermal activity (EDA) and galvanic skin response (GSR) measurement.

14. *“The authors focused their analysis on a very limited number of trials (e.g., late phase acquisition vs. last trial of extinction; last trial extinction vs. first trials reinstatement)... However, we would suggest that the manuscripts should provide a rationale for these choices and indicates whetehr they are common and accepted in the pertinent literature.”*

We thank the reviewer for this suggestion and have added the following sentence to the manuscript:

“We selected the last trial of extinction and the first trial of test phases due to the rapid adaptation of fear responses documented in the relevant literatures and it was a common practice to focus analysis on a limited number of trials^{7,10,32,34,35,36,37,38,39,40,41}.”

15. *“...The authors could clarify on the instructions. What were the verbatim instructions? Where participants asked to suppress the sensation or their knowledge about the aversive history associated with a CS? Was there any practice?”*

On day 2, before the extinction phase, participants were given the following instructions (in Chinese) before they started the active suppression or thought diversion together with extinction training and there was no extra practice before the experiment. In the active suppression experiment, participants were basically asked to suppress their conscious knowledge about the aversive history associated with a CS+.

Active suppression experiment: “In the following task, at the beginning of each trial, there is a white cross presented at the center of the screen. After that, you will either see a ‘No Think’ instruction displayed on the screen or a blank screen, followed by a color square that you saw yesterday. If you see the ‘No Think’ instruction, you should try to actively avoid thinking about the upcoming consequence associated with the color square. You should blank your mind and keep fixated on the color square. If you accidentally think about consequence associated with the color square, you should immediately push that thought out of your consciousness. Try your best to maintain this process until the color square disappears. And when you see a white cross, you should stop this process. However, if you see a blank screen, you should not engage this process and instead observe the association between the upcoming color squares and their consequences.”

Thought diversion experiment: “In the following task, at the beginning of each trial, there is a white cross presented at the center of the screen. After that, you will either see an ‘Imagine’ instruction displayed on the screen or a blank screen, followed by a color square that you saw yesterday. If you see the ‘Imagine’ instruction, you should relate the color square you see to a soothing natural scene that you can come up with. For example, when you see a blue square, you can imagine the serene blue sky; when you see a yellow square, you can relate that to the golden canola flowers. Try your best to maintain this process and fixate on the color square until the color square disappears. And when you see a white cross, you should stop this process. However, if you see a blank screen, you should not engage this process and instead observe the association

between the upcoming color squares and their consequences.”

These instructions were provided to the participants before the experiment and were not included in the main text of the manuscript.

16. *“... it would be helpful ... if the data were published in a neutral format (e.g., .csv) that does not require specific proprietary software...if they were accompanied by a simple read_me file that explains data format, variables and abbreviation... to reproduce the analyses, the authors may consider also uploading the SPSS syntax.”*

We have prepared the data files in the neutral format (.csv) and included a read_me file to explain the details about the data format, trial sequence etc. Also, we have uploaded the SPSS syntax.

17. *“line 122: For readers not familiar with conditioning, a brief explanation of CS1 and CS2 might be helpful.”*

We have added the following sentences to explain CS1 and CS2 in the manuscript:

“In our experiments, participants were trained to associate two colored squares on the computer screen (conditioned stimuli (CS+)) with electric shocks in the acquisition phase (i.e., CS1+ and CS2+). During extinction phase, active suppression (experiment 1) or thought diversion (experiment 2) was applied to one of the CS+ and fear responses to both CS+ were examined in the test phase to test the effects of different mnemonic control methods.”

18. *“When discussing evidence for suppression-induced forgetting and it’s possible impairment in people with low control ability, the authors may wish to discuss the recent meta-analysis by Stramaccia et al. and the recent article by Mary et al. about the Paris terrorist attack.”*

We thank the reviewer for pointing us to the relevant literatures, and have elaborated on the connections between our experiments and the papers in the discussion section:

“Previous literatures suggest that there is a negative association between suppression induced forgetting (SIF) and thought control ability^{22,23}. Such observation corroborates well with reported SIF impairment in PTSD or anxiety patients, whom are characterized by the hallmark of deficient memory and intrusive thought control²⁴. Our results resonate with these findings and further show that fear reinstatement impairment caused by active suppression is also sensitive to the thought control ability whereas thought diversion induced SIF is not (Fig. 3).”

Reference

1. Schiller, D., Monfils, M., Raio, C. M., LeDoux, J. E., Johnson, D. C., & Phelps, E. A. (2010). Preventing the return of fear in humans using reconsolidation update mechanisms. *Nature*, *463*(7277), 49-53.
2. Schiller, D., Kanen, J. W., LeDoux, J. E., Monfils, M., & Phelps, E. A. (2013). Extinction during reconsolidation of threat memory diminishes prefrontal cortex involvement. *Proceedings of the National Academy of Sciences - PNAS*, *110*(50), 20040-20045.
3. Raio, C. M., Hartley, C. A., Orederu, T. A., Li, J., & Phelps, E. A. (2017). Stress attenuates the flexible updating of aversive value. *Proceedings of the National Academy of Sciences - PNAS*, *114*(42), 11241-11246.
4. Anderson, M. C., & Green, C. (2001). Suppressing unwanted memories by executive control. *Nature*, *410*(6826), 366-369.
5. Anderson, M. C. (2004). Neural systems underlying the suppression of unwanted memories. *Science*, *303*(5655), 232-235.
6. Bergström, Z. M., de Fockert, J. W., & Richardson-Klavehn, A. (2009). ERP and behavioural evidence for direct suppression of unwanted memories. *Neuroimage*, *48*(4), 726-737.
7. Anderson, M. C., Reinholz, J., Kuhl, B. A., & Mayr, U. (2011). Intentional suppression of unwanted memories grows more difficult as we age. *Psychology and Aging*, *26*(2), 397-405.
8. Racsmány, M., Conway, M. A., Keresztes, A., & Krajcsi, A. (2012). Inhibition and interference in the think/no-think task. *Memory & Cognition*, *40*(2), 168-176.
9. Benoit, R., & Anderson, M. (2012). Opposing mechanisms support the voluntary forgetting of unwanted memories. *Neuron*, *76*(2), 450-460.
10. Paz-Alonso, P. M., Bunge, S. A., Anderson, M. C., & Gheiti, S. (2013). Strength of coupling within a mnemonic control network differentiates those who can and cannot suppress memory retrieval. *The Journal of Neuroscience*, *33*(11), 5017-5026.
11. Weller, P. D., Anderson, M. C., Gómez-Ariza, C. J., & Bajo, M. T. (2013). On the status of cue independence as a criterion for memory inhibition: Evidence against the covert blocking hypothesis. *Journal of Experimental Psychology. Learning, Memory, and Cognition*, *39*(4), 1232-1245.
12. Gagnepain, P., Henson, R. N., & Anderson, M. C. (2014). Suppressing unwanted memories reduces their unconscious influence via targeted cortical inhibition. *Proceedings of the National Academy of Sciences - PNAS*, *111*(13), E1310-E1319.

13. Catarino, A., Küpper, C. S., Werner-Seidler, A., Dalgleish, T., & Anderson, M. C. (2015). Failing to forget: Inhibitory-control deficits compromise memory suppression in posttraumatic stress disorder. *Psychological Science, 26*(5), 604-616.
14. Wang, Y., Cao, Z., Zhu, Z., Cai, H., & Wu, Y. (2015). Cue-independent forgetting by intentional suppression – evidence for inhibition as the mechanism of intentional forgetting. *Cognition, 143*, 31-35.
15. Hulbert, J. C., & Anderson, M. C. (2018). What doesn't kill you makes you stronger: Psychological trauma and its relationship to enhanced memory control. *Journal of Experimental Psychology. General, 147*(12), 1931.
16. Kindt, M., Soeter, M., & Vervliet, B. (2009). Beyond extinction: Erasing human fear responses and preventing the return of fear. *Nature Neuroscience, 12*(3), 256-258.
17. Kindt, M., & Soeter, M. (2018). Pharmacologically induced amnesia for learned fear is time and sleep dependent. *Nature Communications, 9*(1), 1316-10.
18. Thompson, A., & Lipp, O. V. (2017). Extinction during reconsolidation eliminates recovery of fear conditioned to fear-irrelevant and fear-relevant stimuli. *Behaviour Research and Therapy, 92*, 1-10.
19. Engen, H. G., & Anderson, M. C. (2018). Memory control: A fundamental mechanism of emotion regulation. *Trends in Cognitive Sciences, 22*(11), 982-995.
20. Noreen, S., & Ridout, N. (2016). Intentional forgetting in dysphoria: Investigating the inhibitory effects of thought substitution using independent cues. *Journal of Behavior Therapy and Experimental Psychiatry, 52*, 110-118.
21. Liu, J., Zhao, L., Xue, Y., Shi, J., Suo, L., & Luo, Y., . . . Lu, L. (2014). An unconditioned stimulus retrieval extinction procedure to prevent the return of fear memory. *Biological Psychiatry, 76*(11), 895-901.
22. Dunsmoor, J. E., Murty, V. P., Davachi, L., & Phelps, E. A. (2015). Emotional learning selectively and retroactively strengthens memories for related events. *Nature, 520*(7547), 345-348.
23. Luo, Y., Xue, Y., Liu, J., Shi, H., Jian, M., Han, Y., . . . Lu, L. (2015). A novel UCS memory retrieval-extinction procedure to inhibit relapse to drug seeking. *Nature Communications, 6*(1), 7675-7675.
24. Dunsmoor, J. E., Otto, A. R., & Phelps, E. A. (2017). Stress promotes generalization of older but not recent threat memories. *Proceedings of the National Academy of Sciences - PNAS, 114*(34), 9218-9223.
25. Dunsmoor, J. E., Kroes, M. C. W., Moscatelli, C. M., Evans, M. D., Davachi, L., & Phelps, E. A. (2018). Event segmentation protects emotional memories from competing experiences encoded close in time. *Nature Human Behaviour, 2*(4), 291-299.
26. Hu, J., Wang, W., Homan, P., Wang, P., Zheng, X., & Schiller, D. (2018). Reminder duration determines threat memory modification in humans. *Scientific Reports, 8*(1), 8848-10.
27. Reddan, M. C., Wager, T. D., & Schiller, D. (2018). Attenuating neural threat expression with imagination. *Neuron, 100*(4), 994-1005.
28. Dunsmoor, J. E., Kroes, M. C. W., Li, J., Daw, N. D., Simpson, H. B., & Phelps, E. A. (2019). Role of human ventromedial prefrontal cortex in learning and recall of enhanced extinction. *The Journal of Neuroscience, 39*(17), 3264-3276.
29. Borgomaneri, S., Battaglia, S., Garofalo, S., Tortora, F., & Pellegrino, G. D. (2020). State-dependent TMS over prefrontal cortex disrupts fear-memory reconsolidation and prevents the return of fear. *Current Biology, 30*, 1-8.

REVIEWERS' COMMENTS:

Reviewer #1 (Remarks to the Author):

The authors have successfully addressed all my concerns, and I appreciate that the current manuscript has improved.

Some parts could benefit from proofreading. For example, "we hypothesized" in the method section on pg 6 (line 127) may better read as "we observed/found, etc." Similarly, "no main effect between experiment" should probably be "no main effect of experiment".

Reviewer #3 (Remarks to the Author):

Review of the revised manuscript "Active suppression prevents the return of threat memory in humans" by Ye Wang, Zijian Zhu, Jingchu Hu, Daniela Schiller, Jian Li

Overall, the authors were very responsive to our questions and comments. The revisions improve the clarity of the manuscript and also make a more compelling case with respect to the robustness of the report effect. We only have a few further comments that we will detail below:

(1) We appreciate the authors' reasons for changing thought suppression to active suppression. We do note though that they stick to the old terminology in the abstract and the figure captions.

Re. thought substitution, we would suggest that the original idea proposed by, e.g., Bergström et al. (e.g., 2009) is not that it involves active suppression plus an additional process of thought diversion. In fact, such an account would make it harder to explain why thought substitution sometimes does not have the same inhibitory after effects as active suppression (i.e., forgetting on an independent probe test.).

However, relevant for the current manuscript, we note that the authors do use the term thought substitution – rather than diversion - after all in the introduction. We thus suggest that there is a need for a greater consistency of these terms throughout the manuscript.

(2) We are still somewhat confused by the power analysis and how it was conducted in G*Power (which seems fairly limited in its options). We understand that the authors chose the median effect size of the three cited studies. However, how did they enter a partial η^2 into the software and what statistical test did they select?

(3) Re. to our previous point (7): The authors confirmed their result with the fuller sample that also included "non-learners". We think that this strengthens their argument and makes it less likely that the results are biased in favor of very anxious participants only. In fact, we would suggest focusing on reporting the whole sample (instead of the selected subsample) throughout the manuscript.

(4) Re. our previous point (8): The authors comprehensively addressed all our questions in the response letter. However, to foster reproducibility of this work, they may want to add the information also to the methods or supplement.

(5) Re. our previous point (11): The data files and syntax were uploaded as rar archive. At least Mac users thus require additional tools to access the files. Maybe the authors would consider uploading single files or a zip archive (if that is more universally accessible)?

(6) Was there also a correlation with the TCAQ or "only" a median-split effect? The former would

provide stronger evidence for an association between the two.

(7) We struggled understanding the inserted red paragraph in the supplementary methods section on page 17. It may benefit from rephrasing.

COMMENTS FROM THE REVIEWERS:

Reviewer #1

1. *“The authors have successfully addressed all my concerns, and I appreciate that the current manuscript has improved. Some parts could benefit from proofreading. For example, “we hypothesized” in the method section on pg. 6 (line 127) may better read as “we observed/found, etc.” Similarly, “no main effect between experiment” should probably be “no main effect of experiment.”*

Following the reviewer’s suggestion, we have made corresponding changes in the main text accordingly.

Reviewer #3

1. *“We appreciate the authors’ reasons for changing thought suppression to active suppression. We do note though that they stick to the old terminology in the abstract and the figure captions.
Re. thought substitution, we would suggest that the original idea proposed by, e.g., Bergström et al. (e.g., 2009) is not that it involves active suppression plus an additional process of thought diversion. In fact, such an account would make it harder to explain why thought substitution sometimes does not have the same inhibitory after effects as active suppression (i.e., forgetting on an independent probe test.).
However, relevant for the current manuscript, we note that the authors do use the term thought substitution – rather than diversion - after all in the introduction. We thus suggest that there is a need for a greater consistency of these terms throughout the manuscript.”*

We went through the manuscript and had thus used active suppression and thought diversion where they were appropriate to achieve better consistency.

2. *“We are still somewhat confused by the power analysis and how it was conducted in G*Power (which seems fairly limited in its options). We understand that the authors chose the median effect size of the three cited studies. However, how did they enter a partial eta 2 into the software and what statistical test did they select?”*

We chose the “ANOVA: repeated measures, within-between interaction” option of G*Power to estimate the sample size based on the partial η^2 ($\eta^2 = 0.19$, $\alpha = 0.05$, $\beta = 0.8$, 2 (experiments) x 2 (phases) x 2 (CS) three-way ANOVA interaction effect), and

used “Determine-direct” options to transform “partial η^2 ” into “effect size f” when estimating sample size. Such a procedure is justified by the fact that “the sample size required to detect an effect size for a three-way interaction is exactly fourfold that required to detect the same effect size of a two-way interaction” (Heo et al., 2010). As a result, the total sample size required for the η^2 (0.19) of the two-way ANOVA (2 (experiments) x 2 (phases)) is 12 and the same effect size of the three-way ANOVA (2 (experiments) x 2 (phases) x 2 (CS)) would yield a sample size of 48.

3. *“Re. to our previous point (7): The authors confirmed their result with the fuller sample that also included “non-learners”. We think that this strengthens their argument and makes it less likely that the results are biased in favor of very anxious participants only. In fact, we would suggest focusing on reporting the whole sample (instead of the selected subsample) throughout the manuscript.”*

We also thought about including the “non-learners” into the results but decided instead to focus on the “learners” since the main aim of the paper is to demonstrate the difference of active suppression and thought diversion on fear learning, extinction and fear reinstatement. Adding the “non-learners”, we think, might obscure the main results and readers might question the specificity of our results. Nevertheless, we reported in the manuscript that by adding the “non-learners”, our main results remained almost identical in the methods section to stress the generality of our finding.

4. *“Re. our previous point (8): The authors comprehensively addressed all our questions in the response letter. However, to foster reproducibility of this work, they may want to add the information also to the methods or supplement.”*

We thank the reviewer’s suggestion and have added this analysis and results into the methods section to rule out the possibility that the higher initial response to CS-response in experiment 1 might cause the difference between experiments 1 & 2. We have added the following sentences into the main text:

“To rule out the possibility that different CS- response baseline might play a role in the results we observed, we conducted a mixed-model 3-way analysis of covariance (ANCOVA) with the within-subjects factors CS+ (CS1+ vs. CS2+, defined as the SCR differences between CS+ and CS-), phase (last trial of extinction vs. first trial of test), between-subjects factor experiment (suppression vs. diversion), and the covariate CS- (the difference between the first test trial and the last extinction trial for CS-). There were significant effects of experiment ($F_{1,52} = 5.118, P = 0.028, \eta^2 = 0.090$) and experiment \times phase interaction ($F_{1,52} = 4.982, P = 0.030, \eta^2 = 0.087$), suggesting active suppression and thought diversion still have different effects on fear reinstatement after including the CS- response as a covariate. More specifically, a two-way repeated measure ANCOVA with the factors of CS+ (CS1+ vs. CS2+, defined as the SCR differences between CS+ and CS-), phase (last trial of extinction vs. first trial of test) and the covariate CS- showed no significant effect of phase ($F_{1,26} = 1.184, P = 0.287, \eta^2 = 0.044$), CS+ ($F_{1,26} = 0.709, P = 0.408, \eta^2 = 0.027$), or their interaction ($F_{1,26} = 0.023, P = 0.882, \eta^2 = 0.001$) in experiment 1. Similar ANCOVA was also conducted on experiment 2 and we found that reinstatement effects were significantly different between CS1+ and CS2+ (phase \times CS+ interaction: $F_{1,25} = 10.025, P = 0.004, \eta^2 = 0.286$). Therefore, our results remain robust after controlling for the individual difference of the CS- level.”

5. *“Re. our previous point (11): The data files and syntax were uploaded as rar archive. At least Mac users thus require additional tools to access the files. Maybe the authors would consider uploading single files or a zip archive (if that is more universally accessible)?”*

We have uploaded all the files separately.

6. *“Was there also a correlation with the TCAQ or “only” a median-split effect? The former would provide stronger evidence for an association between the two.”*

Unfortunately, there was only a median-split effect of TCAQ on fear recovery index.

We found that the level of fear recoveries of both CS1+ and CS2+ were not significantly correlated with the scores of TCAQ in experiments 1 or 2 (CS1+: $r = -0.249, P = 0.20$; CS2+: $r = -0.345, P = 0.073$ for experiment 1; CS1+: $r = 0.164, P = 0.413$; CS2+: $r = 0.093, P = 0.644$ for experiment 2).

7. *“We struggled understanding the inserted red paragraph in the supplementary methods section on page 17. It may benefit from rephrasing.”*

We have rewritten this paragraph as the following:

“The criterion for SCR difference (CS+ vs. CS-) increase during extinction was defined as the conjunction of the following conditions: the SCR difference of the last trial larger than that of the first trial of extinction, and the mean difference of latter half trials larger than that of the first half trials in the extinction, and the mean difference of latter half trials larger than that of the first trial, and the last trial difference larger than the mean difference of the first half trials in the extinction^{7,31,32}.”

Reference

Heo, M., & Leon, A. C. (2010). Sample sizes required to detect two-way and three-way interactions involving slope differences in mixed-effects linear models. *Journal of Biopharmaceutical Statistics*, 20(4), 787-802.